# Application of High-Temperature Copper Diffusion in Surface Recoloring of Faceted Labradorites

Qingchao Zhou [1,2,3,*], Chengsi Wang [1,3] and Andy-Hsitien Shen [1,3,*]

1 Gemmological Institute, China University of Geosciences, Wuhan 430074, China; wangcs@cug.edu.cn
2 School of Jewelry, West Yunnan University of Applied Sciences, Baoshan 679100, China
3 Hubei Province Gem & Jewelry Engineering Technology Research Center, Wuhan 430074, China
* Correspondence: qczhou@wyuas.edu.cn (Q.Z.); shenxt@cug.edu.cn (A.-H.S.)

**Abstract:** Owing to the high market values of natural sunstones in Oregon, a kind of artificially diffused red feldspar exhibited at the Tucson Exhibition at the beginning of this century, whose color origin is the same as that of natural sunstone (copper nanoparticles). However, the details of the artificial diffusion process are less disclosed, there is no systematic method to obtain such gemstones. In this paper, we developed the high-temperature copper diffusion process for the surface recoloring of faceted labradorites, which are partly buried in the diffusant. By optimizing the experimental parameters of high-temperature copper diffusion, we successfully recolored the faceted labradorites to red and light red. The gemological and spectroscopic characteristics of the recolored faceted labradorite were further characterized. The red and light-red faceted labradorites exhibited the unique surface plasmon resonance absorption peaks of copper nanoparticles near 580 nm, which is the origin of red color. The typical inclusions formed in the faceted labradorite is in the shape of "fire cloud". The interface of red and light-red faceted labradorite that is in contact with the diffusant is less contaminated, we believe that the contamination could be further reduced or eliminated by optimizing the high-temperature copper diffusion process. The way that the sample is in contact with the diffusant partly is versatile and promising in the surface treatment of materials that have already been processed.

**Keywords:** labradorite; faceted; copper diffusion; recoloring

## 1. Introduction

Feldspar is one of the most important rock-forming minerals in the earth's crust, and those with superior quality can be used as gemstones. The feldspar gemstones on the market mainly include moonstone, sunstone, amazonite, iridescence labradorite, and other species with special optical effects. Among them, the sunstone shows strong golden and red metallic luster under the light owning to its sheet metal inclusions. Furthermore, there is a special kind of sunstone that not only has the common daylight effect, but also can show the red and green body colors like ruby and emerald at the same time. Because of its special optical characteristics, this type of gems is favored by many gem carving artists and collectors. As the state stone of Oregon, the rarity and beauty make natural sunstones a high market price. In terms of this kind of natural sunstone, the origins that have been confirmed by gemmologists are Oregon in the United States and Afar region in Ethiopia [1–3].

The research on the coloring mechanism of natural sunstone has never stopped since the discovery of this kind of gemstone in 1908. However, due to the technical limitations, the idea of the existence of copper nanoparticles in natural sunstone has not been directly verified for more than 30 years since then [4–6]. Until 2019, Wang et al. from our research group directly observed the microscopic morphology of copper nanoparticles in natural sunstone by means of FIB-TEM [7]. This problem that has puzzled people for more than 100 years was finally solved.

It is also because of the high price and huge market potential of natural sunstones that an artificially diffused red andesine feldspar was unveiled at the Tucson Exhibition in the United States at the beginning of this century [8–10]. The gem sellers claimed that the red andesine feldspar is natural from a new origin, which aroused the attention of many gem research institutes all over the world. Since the diffused red andesine feldspar has the same coloring mechanism (copper nanoparticles) with natural sunstone, it took nearly ten years of exploration to confirm that this kind of "new origin red andesine feldspar" is a scam [11–16].

Presently, except for the beryllium diffusion of sapphire, most of the diffusion treatment for recoloring is limited to the shallow surface of the gemstone. Nevertheless, in the process of high-temperature diffusion treatment, the diffusant will inevitably cause contamination or damage to the gemstone surface, which need to be removed by a secondary polishing process [17]. The secondary polishing will also remove the colored areas on the surface that are introduced by diffusion. In our previous study, we have proposed a "partly burying" method for the high-temperature copper diffusion of labradorite and andesine feldspars, and systematically investigated the $Cu^+$-$Na^+$ ions exchange and in situ formation process of copper nanoparticles in labradorite [18]. By using the "partly burying" method, we can bury only a part of the labradorite in the diffusant, and the surface of labradorite that is not in contact with the diffusant can avoid the contaminations.

In this paper, we further explored the application of high-temperature copper diffusion in surface recoloring of faceted labradorites, the inconspicuous pavilion of the faceted labradorite was preferably adopted as the partly burying areas in diffusant. The red and light red faceted labradorites were prepared by adjusting the parameters of high-temperature copper diffusion. In addition, we characterized the absorption spectra of the red and light red faceted labradorites to evaluate the color changes, all the samples exhibit unique surface plasmon resonance (SPR) absorption peaks of copper nanoparticles near 580 nm. The "fire cloud" like inclusions and minor surface contaminations of the faceted labradorite after copper diffusion were systematically characterized by microscopic examinations. The "partly burying" method proposed for the surface recoloring of faceted labradorites in this paper may also have great application prospects in other faceted gemstones.

## 2. Materials and Methods

### 2.1. Materials

The labradorite crystals were purchased from Oregon mine, which is available on their official website. All reagents were used as received without further purification: $ZrO_2$ (zirconium (IV) oxide 99%, Aladdin), CuO (copper (II) oxide 99%, Aladdin), activated carbon (C 98%, Aladdin).

### 2.2. Methods

2.2.1. High-Temperature Copper Diffusion of Faceted Labradorites

Six labradorite rough stones from Oregon were selected for cutting and polishing. The information of the six faceted labradorite gems is listed in Table 1.

**Table 1.** Information of faceted labradorites before and after copper diffusion.

| Sample Number and Description | Images before Copper Diffusion | Images after Copper Diffusion |
| --- | --- | --- |
| CP-1<br>oval cut, 2.08 ct<br>10.5 × 7.2 × 4.3 mm | 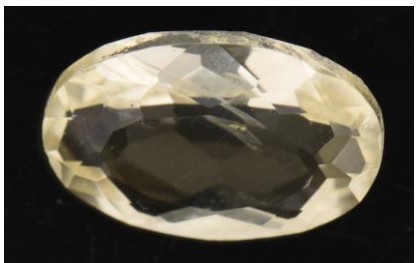<br>pale yellow | 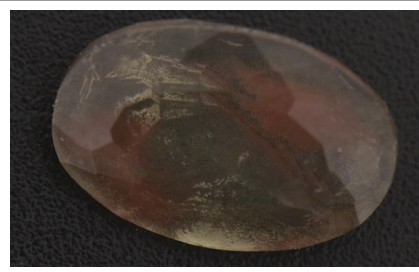<br>gray black (partly red) |

**Table 1.** *Cont.*

| Sample Number and Description | Images before Copper Diffusion | Images after Copper Diffusion |
| --- | --- | --- |
| CP-2<br>oval cut, 2.09 ct<br>10.4 × 7.6 × 4.1 mm | <br>pale yellow | <br>light red |
| CP-3<br>emerald cut, 1.94 ct<br>11.7 × 5.5 × 3.9 mm | <br>pale yellow | <br>light red |
| CP-4<br>pear cut, 2.55 ct<br>11.3 × 8.2 × 5.2 mm | <br>pale yellow | <br>light red |
| CP-5<br>pear cut, 2.77 ct<br>12.6 × 7.4 × 5.7 mm | <br>pale yellow | <br>red |
| CP-6<br>cushion cut, 1.93 ct<br>8.9 × 7.0 × 4.0 mm | <br>pale yellow | <br>light red |

CP-1: Weigh 0.05 g of CuO powder and 5 g of zirconia powder as diffusant, which is ground with an agate mortar to mix uniformly. The faceted labradorite CP-1 is planted in the diffusant. The heating program of the tube furnace for copper diffusion was set at 1050 °C for 6 h and 1150 °C for 3 h, the heating and cooling rate of the tube furnace is 6 °C/min.

CP-2 and CP-3: Weigh 0.05 g of CuO powder and 5 g of zirconia powder as diffusant, which is ground with an agate mortar to mix uniformly. The faceted labradorites CP-2 and CP-3 are planted in the diffusant. An additional 5 g of activated carbon was added to the tube furnace. The heating program of the tube furnace for copper diffusion was set at 1050 °C for 6 h and 1150 °C for 3 h, the heating and cooling rate of the tube furnace is 6 °C/min.

CP-4: Weigh 0.025 g of CuO powder and 5 g of zirconia powder as diffusant, other conditions are the same as CP-2.

CP-5: Weigh 0.1 g of CuO powder and 5 g of zirconia powder as diffusant, the heating program of the tube furnace for copper diffusion was set at 1050 °C for 6 h and 1150 °C for 6 h, other conditions are the same as CP-2.

CP-6: Weigh 0.01 g of CuO powder and 5 g of zirconia powder as diffusant, other conditions are the same as CP-2.

### 2.2.2. Characterizations and Measurements

Images of all the samples were captured in a light box (D55 light source) under identical conditions to compare color changes. All the samples were tested for fluorescence reactions under long-wave (365 nm) ultraviolet (LW-UV). Microscopic observations were performed with a Leica M205A. Ultraviolet-visible (UV-Vis) absorption spectra were characterized by using a Lambda 650S spectrometer. The wavelength covers the ultraviolet to near-infrared band from 300 to 900 nm, the data interval is 1 nm, and the slit width is 5 nm. Fluorescence spectra were characterized by using a Jasco FP8500 fluorescence spectrometer. The photomultiplier tube (PMT) voltage was fixed at 600 V for all samples to compare the fluorescence intensity. The emission spectra (340–750 nm) was measured with the excitation wavelength of 320 nm at a response time of 0.5 s and a scan speed of 1000 nm/min. For all samples, the parameter settings of the instrument remain unchanged. Raman spectra were characterized by using a micro confocal Raman spectrometer (HORIBA LabRAM HR Evolution) in air using a 532 nm laser, 6.25 mW of laser power, and 9–15 $cm^{-1}$ resolution. The light was focused on an approximately 2 μm spot. Spectra were collected in the wavenumber range of 45–4450 $cm^{-1}$. Major and trace element analyses were conducted by laser ablation inductively coupled plasma mass spectrometry (LA-ICP-MS) at the State Key Laboratory of Geological Processes and Mineral Resources, China University of Geosciences, Wuhan. Detailed operating conditions for the laser ablation system and the ICP-MS instrument and data reduction are the same as described by Liu et al. [19]. All data were acquired on sample in single spot ablation mode at a spot size of 44 μm in this paper. Each analysis incorporated a background acquisition of approximately 20–30 s (gas blank) followed by 50 s of data acquisition from the sample. The Agilent Chemstation was utilized for the acquisition of each individual analysis. Element contents were calibrated against multiple-reference materials (BCR-2G, BIR-1G and BHVO-2G) without applying internal standardization.

## 3. Results and Discussion

### 3.1. High-Temperature Copper Diffusion of Faceted Labradorites

Figure 1 illustrates the copper diffusion of faceted labradorite, which is partly buried in the diffusant. The labradorite was first cut and polished to nearly colorless and transparent faceted gemstones. Then, the diffusant was prepared by uniformly mixing the CuO and $ZrO_2$ powder and then transferred to an alumina crucible. $ZrO_2$ is selected as the dispersant of CuO because of its high melting point of ~2700 °C, and it hardly reacts with CuO and labradorite crystals during the high-temperature diffusion process. Finally, the pavilion

of faceted labradorite was planted in the diffusant, the alumina crucible planted with the faceted labradorite was placed in a tube furnace for high-temperature copper diffusion experiments.

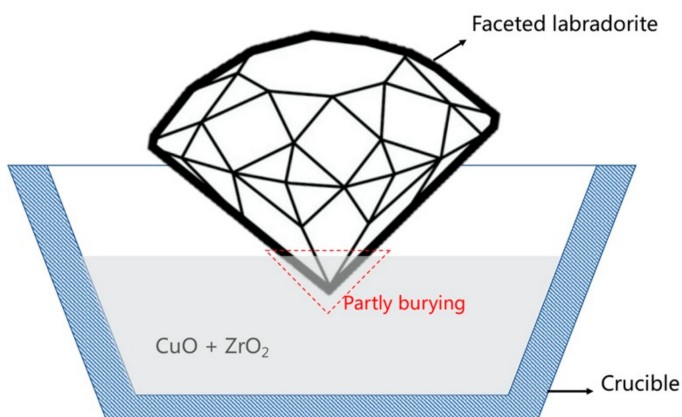

**Figure 1.** Schematic illustration of the copper diffusion of faceted labradorites.

*3.2. Faceted Labradorites before Copper Diffusion*

3.2.1. Gemological Characteristics

Table 1 lists the pictures of the six faceted labradorites that have been cut and polished. Compared with the original rough stones, the color effect of the faceted labradorites has been greatly improved. The rough labradorites used for cutting are not large, so the size of the six faceted labradorite is also relatively small (1.93–2.77 ct). In order to maximize the weight of faceted labradorites, it can be seen from the pictures that many obvious defects (inclusions and cracks) were not avoided during the cutting process. As shown in Figure 2, some typical inclusions can be observed in the six faceted labradorites, such as the "fried egg"-like inclusions, yellow dotted inclusions, white flocs, and healing fissures. It should be noted that these obvious inclusions and healing fissures should be avoided in the cutting process if it is to be processed into commercial grade faceted labradorite.

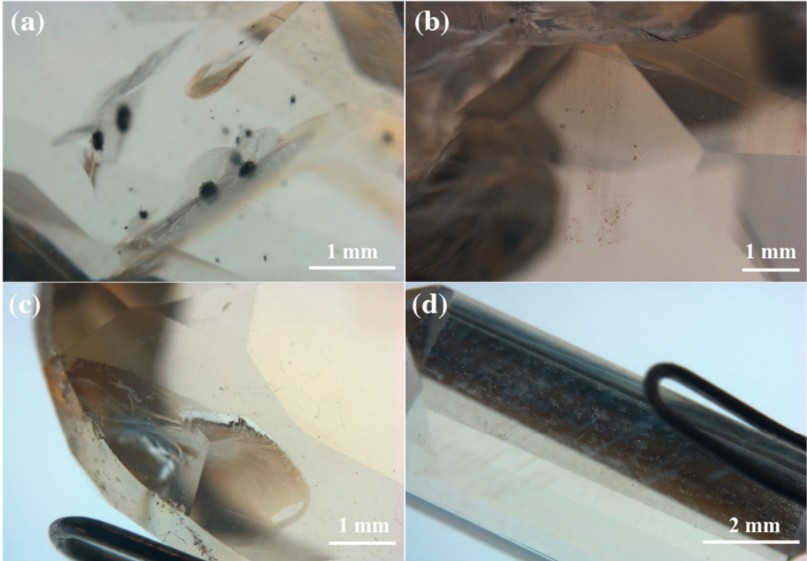

**Figure 2.** Typical inclusions in the faceted labradorites before high-temperature copper diffusion, (**a**) "fried egg"-like inclusions, (**b**) yellow dotted inclusions, (**c**) healing fissures, and (**d**) white flocs.

3.2.2. Spectroscopic Characteristics

Figure 3a shows the absorption spectra of the above six faceted labradorites before copper diffusion. Due to the different thickness of the faceted labradorites, the horizontal

baseline of the absorption spectra is different, but all the faceted labradorites exhibit three absorption peaks at 381 nm, 421 nm, and 447 nm. These three absorption peaks are attributed to the inherent $Fe^{3+}$ ions in natural labradorites [5]. According to the LA-ICP-MS data, the average concentration of element Fe in labradorite is 0.44 wt% (calculated as FeO).

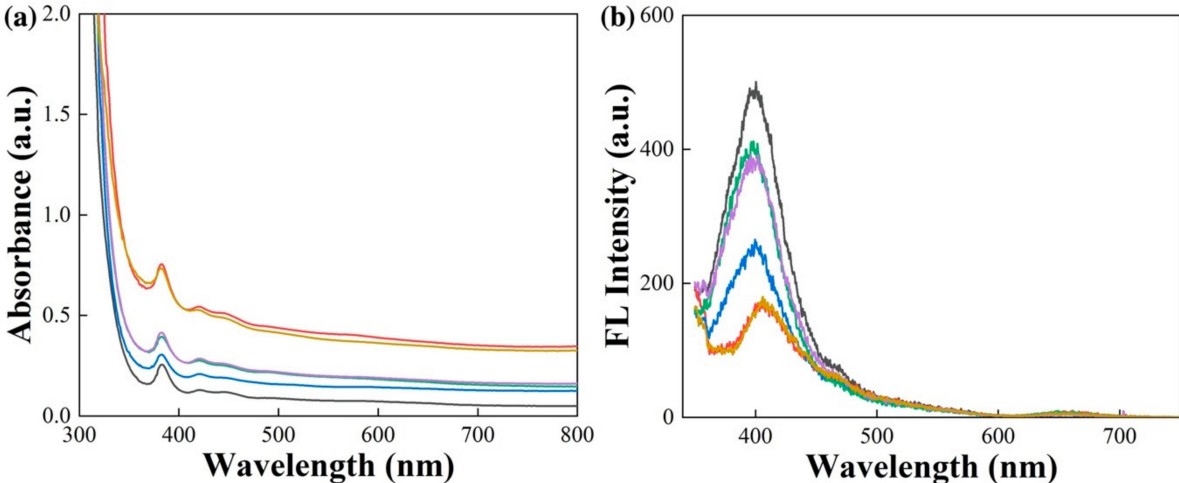

**Figure 3.** UV-Vis absorption spectra (**a**) and fluorescence emission spectra (**b**) of the faceted labradorites.

Figure 3b shows the fluorescence emission spectra of the above six faceted labradorites before copper diffusion. It is obvious that all the six faceted labradorites exhibit a weak emission peak (100–500 cps) around 400 nm under the excitation of 320 nm. We are currently unsure of the attribution of this fluorescence emission peak but tend to believe that it is caused by $Cu^+$, because the labradorite with more $Cu^+$ after high-temperature copper diffusion exhibits a strong fluorescence emission peak near this wavelength. The concentration of copper in the untreated labradorite is 1–10 ppmw determined by LA-ICP-MS.

### 3.3. Recolored Faceted Labradorites by High-Temperature Copper Diffusion

### 3.3.1. The Parameters of High-Temperature Copper Diffusion

The surface of the faceted labradorite was recolored by high-temperature copper diffusion. In order to have a more direct comparison with the sample before copper diffusion, the images of the faceted labradorite after high-temperature copper diffusion are summarized in Table 1. Among them, the sample CP-1 was treated in the diffusant with CuO mass concentration of 1% for 9 h (1050 °C for 6 h + 1150 °C for 3 h). It can be seen from the picture that the sample CP-1 is only turned red partly after the copper diffusion, most areas of the sample turned gray black. In our previous research, the surface of the entire rough labradorite can be treated to red by using this copper diffusion scheme [18]. It can be concluded that the surface of the faceted labradorite is different from the natural rough labradorite, and the surface of the natural rough labradorite is obviously more favorable for the formation of copper nanoparticles during high-temperature copper diffusion. The possible reason is that the surface of the natural rough labradorite corresponds to the relatively fragile surface of the crystal. During the entire process of the rough labradorite from the formation to the mining, it will be subjected to a variety of external forces. Under the action of external forces, the relatively fragile parts of the natural labradorite will break easily. From the perspective of crystal structure, there are more defects on these relatively fragile surfaces. On the surface with more defects, the diffusion rate of $Cu^+$ ions will be greatly accelerated during the high-temperature copper diffusion, which explains the reason why copper nanoparticles are more likely to be formed on the natural surface of labradorite instead of the artificial facets.

In order to recolor the surface of the faceted labradorites, we further added a certain quality of activated carbon to the tube furnace during the high-temperature copper diffusion, which will accelerate the formation of copper nanoparticles. During the high-

temperature copper diffusion, the addition of activated carbon can consume the oxygen in the tube furnace, thereby promoting the transformation of CuO to $Cu_2O$ at high temperatures [20]. This transition increases the concentration of $Cu^+$ ions in the diffusant, which facilitates the subsequent process of $Cu^+$ diffusion and copper nanoparticle formation.

As shown in the pictures in Table 1, the surfaces of sample CP-2 and CP-3 treated by high-temperature copper diffusion turned to light red after adding the activated carbon to the tube furnace. The purpose of adding two samples is to verify the reproducibility of the copper diffusion, and the influence of small differences among different samples can be also excluded. Moreover, the contamination and damage of faceted labradorites caused by diffusant can be greatly reduced after adding activated carbon to the tube furnace.

In addition, we further recolor the faceted labradorite into different shades of red by adjusting the CuO concentration in the diffusant during the high-temperature copper diffusion. As shown in the pictures in Table 1, when the mass concentration of CuO in the diffusant is reduced (0.2–0.5%), the color of the faceted labradorites CP-4 and CP-6 treated by copper diffusion becomes lighter, when the mass concentration of CuO in the diffusant is increased (2%), the color of the faceted labradorite CP-5 treated by copper diffusion becomes darker. The copper concentration of the faceted labradorites treated by high-temperature copper diffusion was tested by LA-ICP-MS. The copper concentration on the shallow surface of the CP-5 sample with the darkest color was 2780 ppm. The copper concentration on the shallow surface of samples CP-4, CP-6, CP-2, and CP-3 were 451, 329, 893, and 878 ppm, respectively. The above results show that more copper can be incorporated to the shallow surface of the labradorite by increasing the concentration of CuO in the diffusant.

### 3.3.2. Gemological Characteristics of the Recolored Faceted Labradorites

The recolored faceted labradorites were first studied by using classical gemological equipment and methods, namely, the refractometer, longwave ultraviolet light (LW-UV) and microscope. All samples showed more or less equal values for refractive indices, birefringence, and very weak pleochroism (Table 2). Under LW-UV, the pale red faceted labradorites showed weak orange fluorescence, and the red stone (CP-5) showed relatively stronger orange fluorescence. The gemological properties of some labradorites and andesines reported in other references are listed in Table 2 as a comparison.

**Table 2.** Comparison of gemological properties of labradorite and andesine reported in other references [8,9,13].

| | Treated Labradorite from Oregon | Sunstone from Oregon | Andesine from Congo | Labradorite from Congo | Andesine from Tibet |
|---|---|---|---|---|---|
| RI | this paper RIα 1.561–1.563 RIβ 1.570–1.572 | Krzemnicki, 2004 [9] RIα 1.560–1.565 RIβ 1.570–1.572 | Fritsch, 2002 [8] RIα 1.551 RIβ 1.560 | Krzemnicki, 2004 [9] RIα 1.553–1.555 RIβ 1.562–1.563 | Abduriyim, 2009 [13] RIα 1.550–1.552 RIβ 1.555–1.557 |
| DR | 0.009–0.010 | 0.007–0.010 | 0.009 | 0.007–0.011 | 0.009–0.010 |
| color | pale red and red | pale red and pale green | red | red and green | reddish orange, orange-red, and deep red |
| pleochroism | very weak | very weak (red sample) to distinct (green sample) | very weak | very weak (red sample) to distinct (green sample) | weak |
| transparency | transparent | transparent | transparent | transparent | transparent to translucent |
| LW-UV | weak to orange | none | none | weak to distinct orange | weak chalky orange |
| observations | "fire cloud" like inclusions, tiny particles with metallic luster | slight "schiller" effect | "schiller" effect | milky turbidity green labradorite: red under incandescent light | color zoning, twin lamellae, turbid milky clouds and particles, lath-like hollow channels, pipe-like growth tubes, irregular dislocations, fractures, uncommon tiny native copper grains or platelets |

### 3.3.3. Optical Microscopic Characteristics of the Recolored Faceted Labradorites

The microscopic characteristics of the faceted labradorite after high-temperature copper diffusion were analyzed by optical microscopy, including the contaminations in the contact interface between faceted labradorite and the diffusant, the inclusions distributed in the shallow surface of faceted labradorite. The microscopic characteristics and relevant pictures are summarized in Table 3.

**Table 3.** Microscopic images of recolored faceted labradorites by high-temperature copper diffusion.

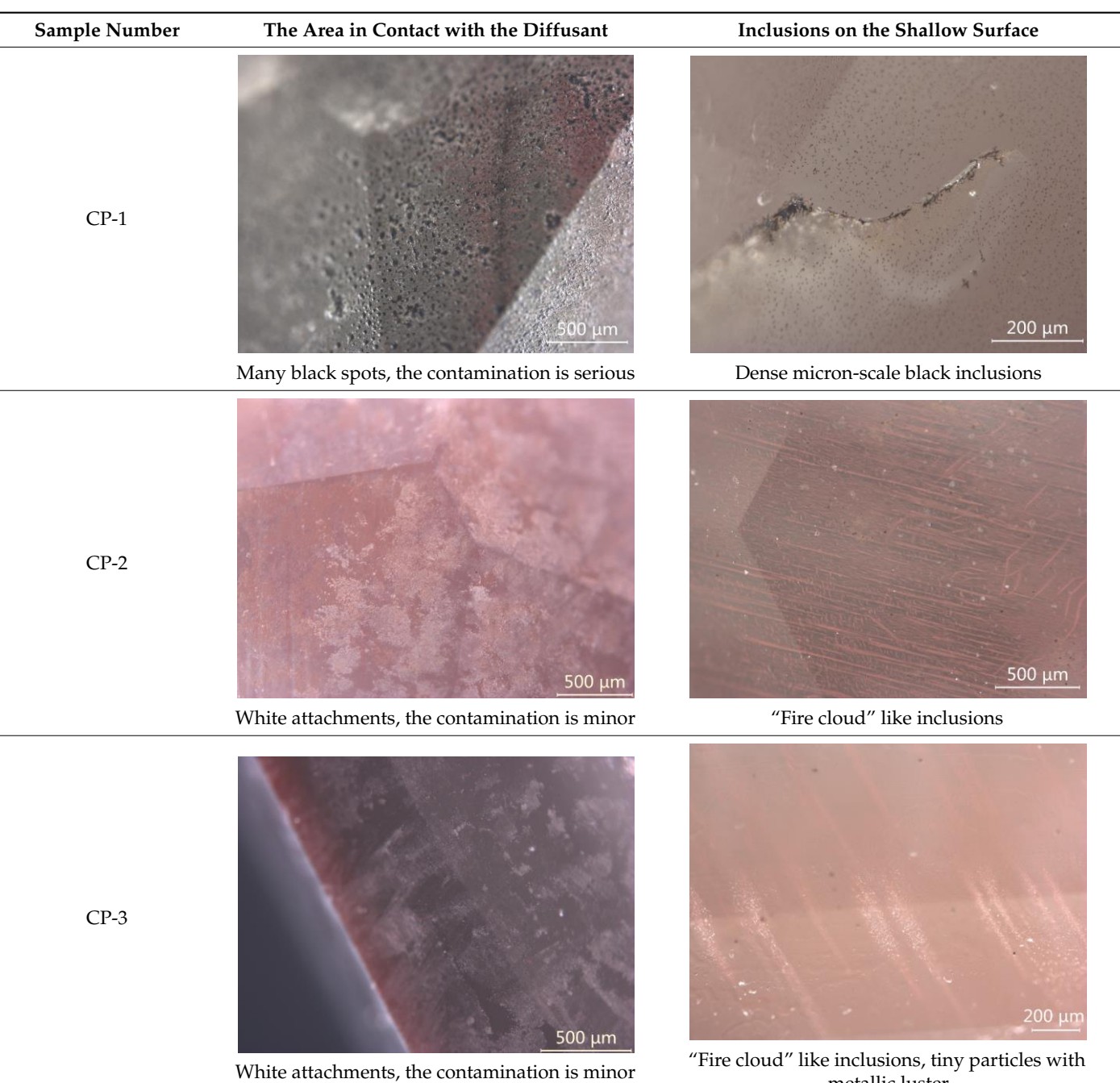

| Sample Number | The Area in Contact with the Diffusant | Inclusions on the Shallow Surface |
|---|---|---|
| CP-1 | Many black spots, the contamination is serious | Dense micron-scale black inclusions |
| CP-2 | White attachments, the contamination is minor | "Fire cloud" like inclusions |
| CP-3 | White attachments, the contamination is minor | "Fire cloud" like inclusions, tiny particles with metallic luster |

**Table 3.** *Cont.*

| Sample Number | The Area in Contact with the Diffusant | Inclusions on the Shallow Surface |
|---|---|---|
| CP-4 |  White attachments, the contamination is minor |  "Fire cloud" like inclusions |
| CP-5 |  Few white attachments, and the contamination is very slight |  "Fire cloud" like inclusions |
| CP-6 |  White attachments, the contamination is minor |  "Fire cloud" like inclusions, tiny particles with metallic luster |

It can be seen from the picture in Table 3 that serious contamination (many black spots) was present in the surface area of the recolored labradorite where the CP-1 sample was in contact with the diffusant. Emmett analyzed that this phenomenon is due to the reaction of CuO in the diffusant with the oxide components (CaO, $SiO_2$, etc.) on the surface of the labradorite to form a solid solution [17]. In the absence of activated carbon in the tube furnace, there will be a large amount of CuO in the diffusant at high temperature, which is the main reason for the contamination of labradorite during the copper diffusion. Moreover, there are many black dots distributed in the shallow surface area that are not in contact with the diffusant. By measuring under an optical microscope, the size of these tiny black particles is 1–6 μm, most of them are tiny rods, the short-axis direction is 1–3 μm,

and the long-axis direction is 3–6 μm. The gray black color of the faceted labradorite after copper diffusion is attributed to these micron-scale inclusions, which then characterized them by micro-Raman spectroscopy. As shown in Figure 4, the Raman spectrum of these black micron-scale inclusions matches that of CuO, appearing Raman peaks at 291, 338, 623, and 1100 cm$^{-1}$. We inferred that these micron-scale CuO particles are aggregated by the Cu$^{2+}$ diffused to the shallow surface of the labradorite.

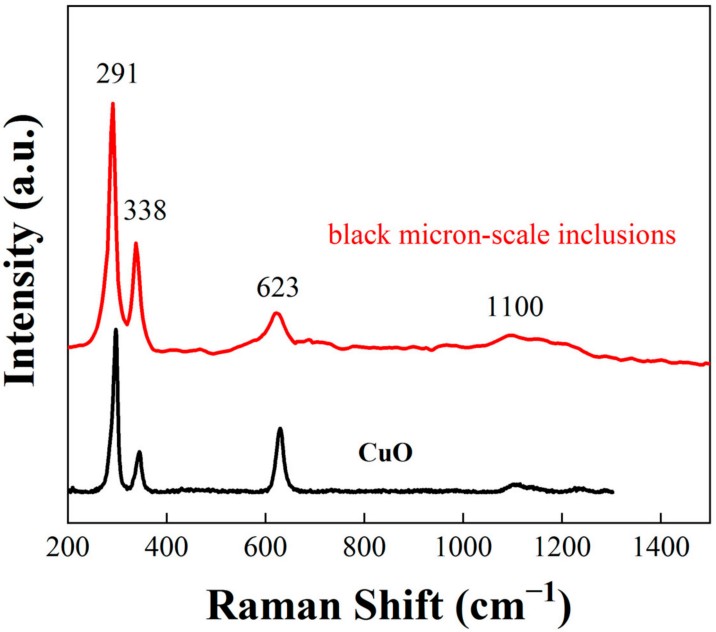

**Figure 4.** Raman spectra of black micron-scale inclusions in recolored faceted labradorite CP-1 and standard CuO (532 nm laser excitation).

To avoid the formation of black micron-scale CuO inclusions in the labradorite, it is necessary to find a way to reduce the concentration of CuO in the diffusant. In our previous study, we have analyzed the thermodynamic reasons for the stable existence of CuO in the diffusant in air atmosphere. In order to improve the conversion efficiency of CuO to Cu$_2$O at high temperature over 1050 °C, the most effective way is to reduce the oxygen fugacity in the reaction conditions, that is, adding substances that can consume oxygen in the tube furnace. In this way, the transformation rate of CuO→Cu$_2$O is increased as soon as the temperature reaches 1050 °C. On the one hand, the transformation of CuO to Cu$_2$O can reduce the reaction of CuO in the diffusant with the labradorite so as to reduce the contamination on the surface. On the other hand, the transformation of CuO to Cu$_2$O can increase the concentration of Cu$^+$ in the diffusant so as to promote the subsequent formation of copper nanoparticles on the shallow surface of labradorite. In addition, we maintained the temperature at 1050 °C for 6 h during the copper diffusion experiment, at which the reaction rate of CuO with labradorite is low but the transformation of CuO→Cu$_2$O had already begun.

From the pictures in Table 3, it can be seen that black micron-scale inclusions are no longer observed on the shallow surface of samples CP-2 and CP-3, which indicates that the addition of activated carbon accelerates the transformation of CuO→Cu$_2$O in the diffusant. The copper in the diffusant mostly exists in the form of Cu$_2$O. During the subsequent high-temperature copper diffusion, the Cu$^+$ diffused into the labradorite is further reduced to Cu$^0$ and aggregated to form copper nanoparticles. The red inclusions formed on the shallow surface of labradorite is in the shape of "fire cloud". The red inclusions of sample CP-3 are further enlarged to show tiny particles with metallic luster, which are supposed to be the copper particles. In addition, the interfaces of the two labradorites that are in contact with the diffusant no longer have black spots contaminations, but some white attachments.

This kind of surface contamination should be further eliminated by optimization of the copper diffusion process.

By decreasing the concentration of CuO in the diffusant, we prepared the light red samples CP-4 and CP-6 by copper diffusion. As can be seen from the pictures listed in the Table 3, the surface contamination of these two samples is comparable to that of CP-2 and CP-3. The light red in color corresponds to the less "fire cloud" like red inclusions. Similarly, by increasing the concentration of CuO in the diffusant, we prepared the dark red sample CP-5 by copper diffusion. The surface contamination of CP-5 was the least among all samples. Except for a small area, the whole surface of the faceted labradorite is covered with "fire cloud" like red inclusions. The copper diffusion scheme revealed that properly increasing the concentration of CuO in the diffusant can not only make the color of the labradorite darker, but also reduce the surface contamination to a certain extent. In our opinion, the increased concentration of CuO in the diffusant increases the amount of $Cu^+$ at the interface between the diffusant and labradorite, thereby reducing the contact between $ZrO_2$ and labradorite.

### 3.3.4. Spectroscopic Characteristics of the Recolored Faceted Labradorite

Figure 5 shows the UV-Vis absorption spectrum and fluorescence emission spectrum of the recolored faceted labradorite CP-1 by copper diffusion. It can be seen from the figure that the typical SPR absorption peak of copper nanoparticles near 580 nm is very weak, which indicates that few copper nanoparticles were formed on the shallow surface of the faceted labradorite. The transmittance of the recolored faceted labradorite CP-1 decreases due to the contaminations on the surface and the micron-scale CuO formed on the shallow surface. The fluorescence emission peak around 395 nm of the faceted labradorite is also very weak (1400 cps), which indicates that the amount of $Cu^+$ ions diffused into the labradorite is small [21,22]. More copper tends to diffuse into the faceted labradorite in the form of $Cu^{2+}$ instead of $Cu^+$, so the subsequent formation of copper nanoparticles cannot be satisfied.

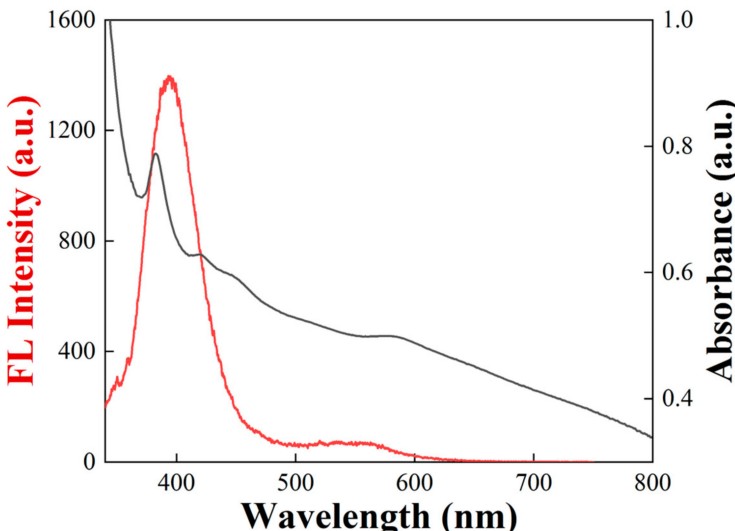

**Figure 5.** UV-Vis absorption (black line) and fluorescence emission spectra (red line) of recolored faceted labradorite CP-1 after high-temperature copper diffusion.

As shown in Figure 6a, the SPR absorption peaks of recolored faceted labradorites are stronger than that of sample CP-1 owing to the addition of activated carbon in tube furnace during high-temperature copper diffusion. The SPR absorption peak of sample CP-5 is the strongest, which corresponds to the most intense red color. From the fluorescence spectra in Figure 6b, it can be seen that the recolored faceted labradorites with the addition of activated carbon exhibit strong fluorescence emission peaks near 395 nm (2700–3500 cps).

The measurement conditions of the fluorescence spectra for all samples in this paper are maintained the same. Presently, it seems that the typical SPR absorption peak of copper nanoparticles and the fluorescence emission peaks of $Cu^+$ appear simultaneously in the recolored faceted labradorite by high-temperature copper diffusion. Moreover, the recolored faceted labradorites by high-temperature copper diffusion are easily distinguishable from the natural Oregon sunstones and will not confuse the consumers. On the one hand, the color of the recolored faceted labradorites is only distributed on the surface; on the other hand, the recolored faceted labradorites exhibit strong fluorescence emission under the excitation of 320 nm ultraviolet light.

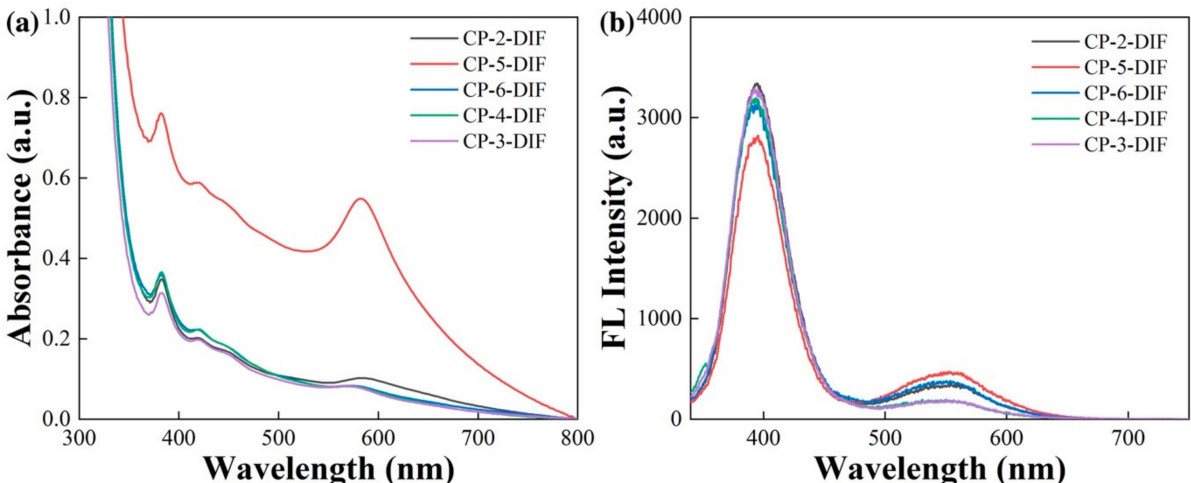

**Figure 6.** UV-Vis absorption spectra (**a**) and fluorescence emission spectra (**b**) of recolored faceted labradorites CP-2, CP-3, CP-4, CP-5, CP-6 by high-temperature copper diffusion.

## 4. Conclusions

Based on the way that the sample is in contact with the diffusant partly, we here successfully recolored six faceted labradorites by using high-temperature copper diffusion. Among them, a faceted labradorite is gray black with a little red, a faceted labradorite is red, and the remaining four faceted labradorites are light red. Spectroscopic data illustrate that the SPR absorption peak of copper nanoparticles near 580 nm can be observed in the UV-Vis absorption spectra of all six faceted labradorites, the gray black one is the weakest and the red one is the strongest. Fluorescence spectral data show that the red and light red faceted labradorites exhibit stronger fluorescence emission of $Cu^+$ near 395 nm than the gray black one. Micro-Raman spectrum shows that the gray black color of the faceted labradorite is attributed to the distribution of dense gray black CuO inclusions (1–3 μm) on the shallow surface. Differently, the typical "fire cloud"-like inclusions are observed on the shallow surface of both red and light red faceted labradorites. The surface contamination of gray black faceted labradorites encountered the diffusant is more serious because of the chemical reaction between CuO in the diffusant and the metal oxide composition in labradorite. By introducing the activated carbon to the tube furnace, the surface contamination of the red and light red faceted labradorites is greatly reduced because the CuO in the diffusant was converted to $Cu_2O$. We believe that the surface contamination could be further reduced or eliminated by optimizing the high-temperature copper diffusion process. More importantly, the way that the sample is in contact with the diffusant partly is versatile to the surface recoloring of other faceted gemstones.

**Author Contributions:** Conceptualization, Q.Z.; methodology, Q.Z.; validation, Q.Z. and C.W.; writing—original draft preparation, Q.Z.; writing—review and editing, A.-H.S. All authors have read and agreed to the published version of the manuscript.

**Funding:** This paper is CIGT contribution CIGTWZ-2022021. The authors acknowledge the financial support of a Grant (CIGTXM-04-S202146) from Center for Innovative Gem Testing Technology, China University of Geosciences (Wuhan). This research was funded by the Special Basic Cooperative Research Programs of Yunnan Provincial Undergraduate Universities' Association (grant NO. 202101BA070001-028).

**Data Availability Statement:** Data is available from the corresponding authors.

**Acknowledgments:** The authors thank Jia Liu for her help in measurements.

**Conflicts of Interest:** The authors declare no conflict of interest.

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
