# Peer review of "Application of High-Temperature Copper Diffusion in Surface Recoloring of Faceted Labradorites"

_minerals, doi:10.3390/min12080920_

Round 1
Reviewer 1 Report
Comments on the manuscript
“Application of High-Temperature Copper Diffusion in Surface Recoloring of Faceted Labradorites”
by Qingchao Zhou, Chengsi Wang and Andy Hsitien Shen
This work is a substantial developed and study of high-temperature copper diffusion process for the surface recoloring of various labradorites which were obtained from Oregon mine. All necessary reagents were used without additional purification: ZrO2 - 99%, CuO - 99%, activated carbon C 98%.
The problem of studying the mechanism of coloring of natural solar stone has existed for many years since its discovery in 1908. The lack of technical capabilities did not allow solving these tasks. The authors managed to overcome these difficulties and successfully resolve this problem.
In their work, the authors successfully use the following widely known spectroscopic methods: ultraviolet-visible absorption, fluorescence, Raman scattering and optical-microscopic methods.
In my opinion, the work is very relevant and has significant practical application.
I think that the article may be of interest both to specialists in this field and to a wide audience of the scientific community.
The entire experiment was carried out at the expert level and the manuscript was written professionally, which prompts me to recommend this article for publication in the journal Minerals.

Author Response
Special thanks to you for your good comments.
Reviewer 2 Report
Dear Authors, indeed the work seems to be very detailed and sounding, however, there are questions that arise after I carefully read your manuscript.
In the first place, what's the reasoning to color the facets of natural gemstones? In the second place, what is the mechanism of diffusing CuO? And how the reduced particles are formed? If Cu(II) is reduced at Cu(I) or CU(0) something else must have oxidized and I don't fully understand what since it seems the atmosphere for the diffusion is air. In third place, Why use CuO diluted in ZrO2? Are the thermal properties of ZrO2 involved in the process? How far from the surface these particles can go? I mean what is the average depth you can find those particles? (I guess you should be able to see them with SEM/TEM/HRTEM measurements. What about defects and particles on defects/grain boundaries (highly more energetic)?
Did you see uniformity in the distribution of the sizes? Did you detect uniformity of distribution on the whole surfaces or the nanoparticles were concentrated more in certain facets?
What is the overall meaning of this process? I mean what are the technological advancements, in practical terms? In which fields can this technology be applied?
Why is the activated carbon used? What is its role?
I think it is not clear what is the take-home message from your article and you should work on it more deeply and answer those questions in the text.
Reviewer 3 Report
A straight-forward presentation of your results.
Author Response
Special thanks to you.